# Assessment of Destructive Impact of Different Factors on Concrete Structures Durability

**DOI:** 10.3390/ma15010225

**Published:** 2021-12-29

**Authors:** Janusz R. Krentowski

**Affiliations:** Faculty of Civil Engineering and Environmental Sciences, Bialystok University of Technology, Wiejska 45E, 15-351 Bialystok, Poland; janusz@delta-av.com.pl

**Keywords:** concrete structures, exceptional load, explosive pressure, structural damage, strengthening

## Abstract

The durability of concrete structure members is dependent on several factors that should be analyzed at each stage of the construction process. Omitting any of these factors might lead to the augmentation of harmful interactions and, as an effect, to safety hazards and the degradation of a structure or its parts. The article, based on several years of studies on exploited concrete structures, presents the effects of an incorrect analysis of selected factors resulting in the occurrence of faults significantly influencing the possibility of safe use of the objects. The described cases include, but are not limited to, the consequences of an improper assessment of building conditions after a biogas explosion in a fermentation chamber, the effect of a wood dust explosion, fire temperature and firefighting action on the prestressed girders, the stages of degradation of bearing structures supporting gas tanks exploited in an aggressive environment, and the consequences of omitting the temperature load in relation to the upper surface of a plate covering the fire pond. In each case, methods of restoration of the damaged elements were proposed, and their application to engineering practice was described. The practical aspects of the conducted research and implemented interventions were indicated.

## 1. Introduction

At a time when effective analytical theories were not known, guidelines for determining safe sections of building elements were obtained from experimental studies [1,2]. Construction failures were also of great importance in the development of strength theories. Conclusions resulting from the practice constituted a source of information for designers and builders. As a result, research on the actual condition of a structure made it possible to develop computational methods for the assessment of the load-bearing capacities of structural elements in order to improve the technology of material production and the technological processes of shaping and erecting structures. These results allowed for the avoidance of errors and flaws in the future, as long as they were correctly interpreted. It was of particular importance in relation to the repetitive phenomena and structures, e.g., precast systems of industrial or residential construction [3,4]. In modern times, in the period of the development of industrial construction and the use of numerical computational techniques, as well as innovative research techniques, such as Digital Image Correlation [5,6], the analysis resulting from the monitoring of operated structures is still the basis for the verification of the actual condition of a structure in comparison with the assumed parameters [7,8,9].

The application of the principles recommended by organizations that develop international standards, such as the American Society for Testing and Materials or the European Standardization Organization, to structures operating in emergency stages is limited. Scientific methods should then be used, and the current strength parameters of basic construction materials should be obtained as a result of scientific research of the actual state of the structure [10].

The correctness of conclusions should be ensured by obtaining reliable data from many independent sources, using various techniques and tools. Such sources may be inspections of the facility, archival documentation, or the monitoring of the deformation of structural elements. Information that has been documented experimentally constitutes the basis for calculations and analyses confirming or refuting the formulated assumptions and research results [11]. Obtaining reliable, i.e., computationally confirmed, results of the conducted experiments makes it possible to carry out further tests by repeating proven procedures and standards in the case of similar building degradation events.

## 2. Factors Generating the Destruction Stage of Structural Materials

The dominant factor generating the degradation stage of concrete materials and structures is insufficient durability of the materials and construction elements used, which are inadequate in terms of the operating conditions of the building. The impact of an aggressive atmospheric environment and the technological conditions in industrial plants, as well as the impact of various exceptional loads, applies to materials built during the investment stage, as well as during renovation, repair, reconstruction, or strengthening of facilities [12,13].

The exploited building and engineering structures are exposed to the influence of exceptional loads that are difficult to predict, e.g., explosion pressure or fire temperature [14,15,16]. The actions taken during the rescue operations and testing of various structures subjected to the temperature of fire differ in the type of equipment used and the type of temporary support structure. However, the actions on the whole object, erected from different construction materials, are usually very similar, and the extent of the damage depends more on the medium that was the fire load. The results of tests carried out during rescue operations after biogas explosions in the fermentation chambers of sewage treatment plants, during the process of securing damaged structures, can be adapted to predict and reduce the possible effects of an explosion in technological equipment [17]. Damaged objects should be immediately protected against the possibility of a progressive catastrophe, that is, against spreading damage to other structural elements [18].

The factors generating the threat stage in each of the analyzed cases are errors made at the stage of designing, shaping, and operating structural elements resulting from the so-called ‘human mistakes’. Regardless of the detected structural defects, exceptional loads, failure to comply with assembly tolerance in the case of prefabricated structures, or poor or inadequate quality of building materials, it is, in each of the analyzed cases, possible to diagnose the lack of proper supervision. It is practically impossible to avoid errors, but it is important to diagnose and eliminate them before the occurrence of signaled or unsignaled damage to buildings. The objects that were tested by the author were located in Poland.

## 3. Durability of Endangered Facilities after Many Years of Operation

The durability of building structures is maintained if, in the expected time of operation, the object meets the assumed requirements in terms of serviceability, load-bearing capacity, and stability without reducing its performance. Fixed spherical gas tanks consist of a technological pressure device, which is a tank, and a support structure. The most common threats to safety are degraded reinforced-concrete supporting structures, because the deformation of the supports results in the disruption of the membrane operation of the tank.

A commonly used solution is the storage of propane-butane gas in spherical pressure tanks with a capacity of 195 m^3^ and 600 m^3^, based on reinforced-concrete supporting structures [19]. The objects examined by the author were erected in the 1990s. Research works concerning their durability and formulating recommendations for the strengthening of damaged elements were carried out over the course of 40 years.

### 3.1. Concrete Supporting Structures for Spherical Tanks

The supporting structure of the tanks with a capacity of 195 m^3^ was shaped as six reinforced-concrete vertical columns, braced with a horizontal reinforced-concrete ring connecting the columns. The foundations under the columns were in the form of a reinforced-concrete foundation slab with a regular hexagonal shape. The supporting pillars were placed around the circumference of a circle (Figure 1a,b). The lower segments of the columns were fixed in the foundations, and their upper parts remained free.

Spherical tanks with a capacity of 600 m^3^ were placed on supports shaped as six supporting trestles inclined at an angle of approximately 6^o^ to the vertical axis, resembling an inverted letter ‘Y’ (Figure 1a,c). The branches of the supporting trestles were placed at the top of the reinforced-concrete base, shaped as a regular dodecagon, located at the ground level. The lower segments of the trestles were articulated with the ring. The ring was stabilized by means of six columns anchored in the lower foundation ring, transmitting the loads to the subsoil. The retaining plates of the tank shell were placed on the column heads in a sliding manner, and the stability of the contact zones was ensured using the action of frictional forces.

### 3.2. Damage to Elements of the Reinforced-Concrete Supporting Structures

After several dozen years of operation, the reinforced-concrete supporting structures were subject to the process of destruction. The degree of damage depended primarily on the initial quality of the elements made, the method of securing the concrete surface, and the aggressiveness of the environment. The methods and quality of the ongoing maintenance, which varied significantly among users, had a significant impact on the scale of the damage. In the tested structures, it was found that the concrete strength corresponded to the C16/20 or even the C12/15 class, which had a significant impact in terms of accelerating the degradation process and reducing durability [20].

The expected durability of reinforced-concrete structures used in specialized construction in the atmosphere of an industrial environment is reduced by corrosion processes caused by the action of soft rainwater. The process of reinforcement corrosion in properly constructed and operated reinforced-concrete structures, operating in an environment with lower aggressiveness classes, should be initiated after about 40 years, but in an environment polluted with aggressive substances such as SO_2_, NO_x_, or Cl^−^, this process is significantly accelerated [21].

During the operation of the carbonized elements, the corrosion products increased in volume and generated tensile stresses and, consequently, concrete scratches and loosening of the cover fragments. The damage, which facilitated the access of aggressive factors from the environment, intensified the corrosion processes, while the corrosion losses reduced the load-bearing capacity of reinforced-concrete elements (Figure 2).

Degraded supporting structures posed a threat to the safety of the tanks. The inclination of the supporting structure of the tank with a capacity of 600 m^3^ at an angle of 6° was the main reason for the weakening of the lower zones of the columns. Incorrectly implemented support zones for the tanks on the column heads did not ensure the correct transfer of the acting loads to the supporting elements.

### 3.3. Elimination of Destruction Processes

Strengthening works, conditioning the safe and durable use of endangered objects, were performed after the endangered structures were excluded from operation. Repair work began by securing some of the structures located in the ground. The lower parts of the reinforced-concrete supports were stabilized by correct connections with the foundations. The degraded reinforced-concrete structure was protected against corrosion by adding reinforcing layers of C25/30 class concrete, reinforced with bars with diameters not exceeding 8 mm. The maximum diameter of the reinforcing bars used was determined by the thickness of the reinforcing layer limited by the elements of technological installations.

The next stage of works was the repair of the damaged surfaces of columns and beams located above the ground level. The obtaining of the conditions of adhesion, strength, and tightness of fillings was ensured by designing the spraying technology, called shotcreting [22,23]. The achieving of the conditions of adhesion, strength, and tightness of the fillings was ensured by using the shotcreting method. The concrete mix was laid in layers by pressure, resulting in a total coating thickness of about 4.5 cm. The concrete mix was made of natural washed aggregate with a granulation of 2–16 mm, and washed sand with a granulation of 0–2 mm, various types of cements and water, as well as mineral additives and chemical admixtures used to plasticize the final product. The influence of the phenomenon of shrinkage was limited using special cements. At the same time, work was carried out to strengthen two opposite columns.

As a result, the existing carbonated concrete substances and exposed reinforcement bars were secured and monolithized (Figure 3). After the shotcrete was made, a monolithic, rigid reinforced-concrete structure was obtained, which also took over bending moments resulting from the leaning of the columns [24]. The final stage of the strengthening works was to protect the connection zones of the support brackets of the steel spherical tanks and the surface of the column heads against mutual displacement in the event of exceptional loads.

## 4. Exceptional Load of Biogas Explosion Pressure

The results of many years of research and the effects of reconstruction work on degraded reinforced-concrete supporting structures allowed the use of similar methods and technologies for repairing damage to the reinforced-concrete fermentation chamber of a sewage treatment plant.

The engineering facilities used in biological sewage treatment plants are designed as structures with a lifetime of at least several dozen years. In the technological process of sewage treatment plants using closed tanks, the so-called biogas, which is a natural result of chemical fermentation processes, is produced. It is an effective energy medium, although in inappropriate conditions of use, it can pose a threat to the life of service workers and cause the destruction of engineering structures, which results from their susceptibility to explosions [25].

Exploitation of specialized structures exposed to the danger of explosion in an aggressive environment of municipal sewage implies the necessity to carry out inspections at intervals that are more frequent than required by the provisions of the construction law for a typical industrial facility. Performing detailed tests makes it possible to check whether the basic structural elements of the tanks meet the standard limit conditions in terms of load-bearing capacity and serviceability. A particularly important condition is to ensure the tightness of the chamber in which the liquid medium is collected. A technological problem is the fact that the tank cannot be used during the time necessary to carry out tests and possible repairs.

### 4.1. Fermentation Chambers

Engineering structures operated in an aggressive technological environment are objects that are particularly exposed to the risk of significant degradation of the quality of construction materials during operation (Figure 4).

The durability of the fermentation chambers is reduced by natural destruction processes. Damage caused by a reduction in the strength parameters of the materials used, short-term exposure to exceptional loads, or an unsignaled change in the static pattern of the structure of the building is significant. Natural wear and tear of materials operated for over 30 years in an aggressive environment, uncontrolled and unidentified at the stage of periodic tests, as well as improperly conducted verification of current material parameters, may generate errors in the assessment of the stage of structural damage, e.g., in terms of the impact of exceptional loads. The above-mentioned factors were the most common causes of the analyzed disasters. The state of the threat was usually intensified by the fact that sewage treatment plants are implemented in the form of a battery of several fermentation chambers cooperating with each other.

### 4.2. Damage to Reinforced-Concrete Elements: Concept of the Strengthening

Problems related to the correct assessment of damaged facilities of a sewage treatment plant are illustrated by an example of a breakdown of a digester with a capacity of 3150 m^3^, caused by an uncontrolled increase in biogas pressure followed by an explosion (Figure 5). The phenomenon of degradation of the reinforced-concrete walls of the chamber was the result of an excessive increase in internal pressure, followed by a biogas explosion. The explosion was initiated by an accidental introduction of a flame into the interior of the tank during renovation works in the area of the adjacent twin chamber, which was temporarily out of service.

As a result of the tests, which were carried out under time pressure from the user, who was eager to restart the technological process as soon as possible, no current parameters of construction materials or internal damage to the reinforced-concrete structure were identified, resulting in, e.g., unsealing of the coatings and changes in the static scheme. After the explosion, the rigid connection of the cylindrical shell with the conical shell turned into a semi-hinge connection. However, the previous structural scheme, i.e., with a rigid connection, was incorrectly adopted for the assessment of the coating condition. As a result, the values of internal forces were underestimated.

The results of numerical calculations verifying the post-disaster stage, which took into account the change in the static scheme, made it possible to determine the actual state of stresses and deformations in the concrete structure. The exceeding of both the ultimate and serviceability limit conditions justified the occurrence of cracks in the monolithic wall structure and the tank cover.

According to the provisions of the law, the structure was subjected to a water test before the facility was put into operation and after repair works were carried out on the damaged conical cover. Defects in the form of leaks found at that time were the basis for a reassessment of the condition of the tank.

After completing supplementary tests on defects and actual material solutions, the state of stress of the structure at the design stage and at the stage of instantaneous explosion pressure load was verified [26]. The influence of cracks in reinforced-concrete walls on the safe operation of the facility intended for the collection of a liquid medium was taken into account. It was shown that in the central part of the walls of the tested chamber, the ultimate and serviceability limit conditions of the reinforced-concrete ring section subjected to exceptional load were not met. The conditions of safe operation of the repaired object were verified, taking into account the change in the static scheme of the structure as a consequence of the explosion. A user-approved concept for restoring the correct durability of the facility was developed, the implementation of which enabled the safe resumption of the technological process. Inside the chamber, a cylindrical shell with a wall thickness of 0.25 m was made and was permanently connected with steel bolts with the existing shell, the thickness of which was 0.80 m.

In the analyzed case, the destruction of the tank’s structural elements was a consequence of an explosion initiated by placing an open flame in the zone of contact with biogas. An additional factor stimulating the explosion was the limitation of the combustion space due to the failures of the safety valve ventilation system.

## 5. Exceptional Load of Biogas Explosion Pressure

Similar phenomena, defects in safety and control devices as well as human errors, resulted in the explosion of wood dust and the destruction of the reinforced-concrete production hall.

The explosion phenomenon, presented in several examples, should be treated as an exceptional load [27]. Uncontrolled pressure increases, with limited space and air access, always result in a dust or gas explosion, even without the initiating factor, which is a flame [28]. In conditions of explosion or fire hazard, the conceptual assumptions for the use of various materials and design solutions, as well as the possible consequences of an incorrect assessment of the condition of the structure that has previously been subject to a failure or a catastrophe, should be carefully considered.

Technological processes in industrial plants, using flammable materials, may constitute a potential explosion hazard. Its consequence, apart from the destruction of devices and equipment, is also damage to buildings or industrial engineering structures [29].

### 5.1. The Isaster of an RC Precast Production Hall

The explosion of technological devices used in wood processing initiated a construction disaster of a prefabricated hall with a reinforced-concrete structure (Figure 6). The facility, consisting of several buildings, was commissioned in the 1980s. As a result of the shock wave, high fire temperature, and then cooling and thermal shock resulting from the extinguishing action, concrete in the structural elements lost its elastic properties. This could have led to the spread of the disaster and the total collapse of the hall. The greatest threat was the loss of adhesion of the prestressing tendons to the concrete, the weakened strength of which could not be classified [30,31]. As a result of the propagation of the shock wave pressure after the explosion, there was damage to brick walls made of autoclaved aerated concrete (AAC) blocks in rooms located in adjacent aisles, separated from each other by fire barriers [32].

By analyzing the guidelines set out in the documentation of the technological process of chipboard production and the data recorded by the devices controlling these processes, the values of the basic parameters of the forming device, which had a significant impact on the occurrence of the explosion, were established. Immediately before the explosion, factory control systems indicated a critically dangerous increase in temperature and pressure inside the machine, but they went unnoticed by technical supervision. Devices designed to ventilate the machine were ineffective during the explosion, which was recorded by the control software. The ventilation systems inside the hall, designed to remove excessively concentrated wood dust, also did not work properly, although the information about the system’s defects was generated by devices controlling the panel production process in a 24-h mode.

The lack of a proper reaction of employees supervising the technological production process and the poor technical condition of fire-fighting devices contributed to an increase in the scope of the damage to building structure elements [33].

After the analysis of all the collected information, it was found that two more explosions occurred in the production hall. The first was caused by the spontaneous combustion of a mixture of wood dust and air supplied to the technological machine with a sharp increase in pressure inside it. The accelerator of the reaction was vapor of formaldehyde resin, which was used in the production of the plates. The self-ignition flame ignited the wood dust accumulated in the hall and triggered a secondary explosion that damaged the structure of the prefabricated reinforced-concrete walls and roof of the building.

### 5.2. The Concept of Rebuilding Damaged Elements

As a result of the tests and calculations carried out, it was found that the implementation of the reinforcements taking over the weight of the damaged prestressed girders was not effective. The girders were qualified for dismantling (Figure 7). The works commenced with the dismantling of the damaged and deformed roof plates. The layers of concrete overlay and the upper surfaces of the roof plates were opened and removed, the reinforcement meshes were cut out, and the previously supported ribs of the plates were dismantled. Subsequently, the prestressing strands in the chords of the lower girders were cut. The supported and stabilized girders were disassembled in batches, breaking up the concrete from the central zone towards the supports. Due to the hazardous condition of the structure of the facility, the disassembly of prestressed elements was carried out under the constant supervision of authorized technical personnel. Ongoing geodetic “monitoring“ of the deformation state of the building’s elements was also carried out. At the same time, during the process of cutting the tensioning tendons, a control process was carried out with the use of a set of strain gauges. Their indications after cutting the fibers were stable, which confirmed the fact that the tendons had sufficient adhesion to the cover layer, despite a drastic decrease in the concrete strength of the prestressed girders. The conducted demolition works are shown in Figure 8.

An important conclusion resulting from the conducted research is the recommendation to avoid prestressed reinforced-concrete structures in buildings exposed to internal or external fire hazards, or those under threat of an explosion. Traditional reinforced-concrete girders or steel truss girders result in minor losses in the event of an explosion. In place of the girders used so far, a light covering made of corrugated sheets based on purlins was designed to transfer the load to the steel girders. Taking into account the assembly conditions, each truss was divided into three segments, joined after stabilization on supports and previously made working platforms. The damaged parts of the gable walls made of aerated concrete blocks were replaced with light curtain walls made of corrugated sheets.

The presented catastrophe could be avoided, or its effects significantly reduced, provided that the technical staff reacted correctly to the information and warnings generated by the automatic control system for the compliance of the technological process parameters and their comparison with the designed reference values. Properly coordinated actions after the explosion and damage to the building structure contributed to the reduction in the degradation process in the zones of subsequent production aisles.

The explosion phenomenon has consequences in the form of an additional load with the temperature of the fire and then cooling of the structure during the firefighting operation.

## 6. Environmental Temperature Load

The environmental temperature load, often neglected at the design stage of reinforced-concrete structures, also leads to the surface degradation phenomenon caused by the appearance of uncontrolled scratches or cracks.

Identification of the structure and morphology of loads in the implemented and operated structures is an important aspect of scientific research. The proper selection of research methods [34], the correct interpretation of research and analysis results, and the selection of appropriate technologies to verify assumptions and conclusions must be documented with the knowledge and experience of designers, contractors, and users of building structures. It is particularly important to take into account the consequences of the impact of exceptional loads in the form of non-static loads, e.g., the influence of temperature on elements of engineering structures, resulting in the emergence of the hazard stage [35].

### 6.1. Underground Fire Pond

The underground fire pond was located in the warehouse area of the shopping center building. The rectangular-shaped reservoir, built in 1990, was made of two chambers, each of which was separated by a structural wall with two rectangular openings enabling the flow of water during emptying. The dimensions of the tank were 17.05 × 19.60 m. The bottom plate was 0.40 m thick, the thickness of the reinforced-concrete external walls and the middle wall was 0.30 m, and the thickness of the longitudinal partition walls supporting the upper plate of the tank and reducing the span of the slab was equal to 0.20 m. The structural elements of the tank were made of C25/30-class concrete and the tightness requirements were marked with the symbol W8.

### 6.2. Destruction Control Processes. Structural Defects

As a result of the visual tests, the presence of irregular scratches of the floor, located along the direction of the main reinforcement of the slab, was confirmed. Cracks with a width of 0.5–2.5 mm were located at intervals of 1.2–2.0 m (Figure 9). The concentration of defects occurred along the central longitudinal wall and in other communication spaces that were exposed to the load, with the presence of goods trolleys varying over time. Detailed research confirmed the hypothesis of the occurrence of scratches and cracks also on the lower surface of the tank plate.

The limit state of deflections was identified on the basis of leveling measurements of the actual deformation of the slab. The greatest displacements were obtained in the destruction zones over the total thickness of the slab. The type and condition of the main reinforcement and separating bars in the top slab structure and the actual concrete strength, i.e., the main parameters limiting the scratching process, were identified in the course of the research work.

The upper plate of the tank absorbed the loads resulting from the fixed weight of the plate, the weight of the pallets with the stored goods, and the weight and movement of the means of transport.

Since the analysis of the ultimate load capacity in the direction determined by the main reinforcement bars did not explain the cracking process, the bending state of the plate strands in the perpendicular direction was analyzed, taking into account the distributed reinforcement. It was justified computationally that the plate strips separated with successive cracks created a state of danger, especially during changes in the load caused by the dynamic impact of means of transportation with the simultaneous process of loading with warehouse materials.

It was found that the flaws in the design of the distribution bars of the tank top plate reinforcement stimulated the failure condition of the structure loaded with the environmental temperature load through roof skylights, which was not taken into account at the design stage. The slab’s cross-section was classified as concrete, as the used reinforcement bars had a cross-section area smaller than required by the standards. The load-bearing capacity of the unreinforced-concrete element turned out to be insufficient, which resulted in the destruction of the concrete structure due to bending moments and the formation of cracks in the slab along a direction parallel to the direction of the main reinforcement.

### 6.3. Strengthening the Structure of the Pond Roof Slab

The underground pond located in the warehouse was made of reinforced concrete. For this reason, it was decided to also perform repair works of reinforced concrete, which involved permanently connecting structural supporting elements and restoring proper safety conditions.

On the basis of the tests and identification of the deformation process, it was found that the upper plate did not lose its elastic properties despite significant damage. The consequence of the research work was the development of the concept of the reinforcing structure, and then the commencement of the reinforcement implementation after emptying two of the four compartments of the tank at the same time. Thanks to this technology, the facility still had a water reserve in case of fire. During the works, the commercial facility was in operation. The reinforced-concrete monolithic structure was made of C35/45 concrete, in the form of a four-span beam, supported in nests, profiled in the extreme gable walls of the tank and on square-section columns. The girders were connected with the upper plate of the tank with an individually designed structure of round spindles, reducing the mutual displacement of the separate plate strips.

A similar reinforcing structure was used by the authors of [36,37], who carried out numerical calculations and then implemented tidal turbine reinforcement. In these cases, the execution of works under the sea was an additional difficulty.

## 7. Discussion

In the cases studied and presented in this paper, proper risk assessment after the occurrence of a disaster and efficiently carried out reconstruction works allows the elimination of the risk of degradation phenomena and ensures the fastest possible launch of broadly understood technological processes [38].

If defects are found, especially those resulting from previous destructive phenomena, the structure of the facility should be secured against the possibility of intensifying the threat stage, taking into account:Designed construction solutions;Current static scheme;Real, implemented design solutions;Completed repair and strengthening works;Actual physical and mechanical parameters of construction materials, including their wear and tear;Variable, operational technological parameters.

Re-admitting a degraded structure to service should be related to the assessment of the possibility of:Restoration of the safe serviceability of the existing structure by repairing and strengthening it, taking into account the cooperation of new structural elements;Checking whether the structure can be loaded in accordance with the intended use, taking into account the possible extension of the service life;Reconstruction of a defective structure, e.g., damaged by external factors, especially random situations in terms of the effects of exceptional actions.

In each case, it is necessary to compare the design solutions with the realized state, even if the facility had been used without failure for several dozen years.

As a final result of the research and design process of solutions protecting degraded structures, it is necessary to indicate to the user the period in which the structure can safely meet the limit state conditions in order to adjust the applied solutions to the planned lifetime of the facility.

However, a rebuilt or reinforced structure should be carefully monitored throughout its entire lifetime. Taking into account the fact that industrial facilities are operated continuously, it is recommended to use mainly non-destructive testing, restricting the number of destructive testing and openings performed. The necessity of incurring additional costs related to the exclusion of an object, or a part of it, from operation for the time of carrying out control tests cannot be a criterion that limits the safety assessment of the structure. The results of control tests, carried out in the periods strictly defined in the recommendations of specialists, will allow for an ongoing assessment of the reliability of building structures. The periods between the dates of examinations may not exceed the periods defined in the applicable regulations.

## 8. Conclusions

The possibilities of using real cases to assess the degradation stages of buildings are limited to the analysis of similar structures, used in the same foundation conditions, climatic zones, technological and environmental operational loads. The use of comparative analysis is justified in scientific research due to the expansion of practical knowledge. It is necessary to compare reliably obtained practical effects with the results of verifiable theoretical and computational analyzes. Consistent results make it possible to achieve a probability of the formulated conclusions to the extent sufficient to assess the state of degradation of materials and the causes of damage, and above all, to allow the facility to be operated safely within the prescribed period.

The method of assessing the state of structure degradation based on experimental tests has limitations due to the possibility of the hypothetical occurrence of random phenomena, locally variable structural and material solutions, or different loads, unidentified in the research process. Only long-term monitoring of a number of facilities in which similar construction solutions were applied, and also operated in similar conditions, allows for the formulation of conclusions addressed to people involved in the investment process, from design, to construction, to the operation of the building.

## Figures and Tables

**Figure 1 materials-15-00225-f001:**
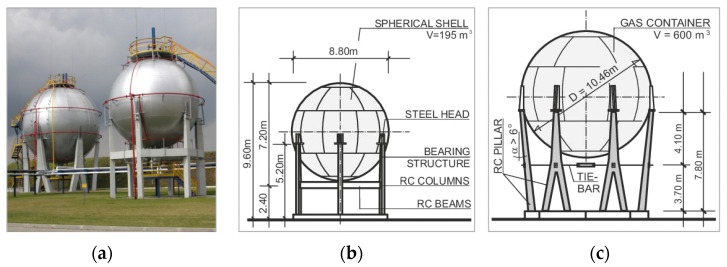
Tanks’ construction details: (**a**) overview of the tanks; (**b**) design details of a 195 m^3^ tank; (**c**) design details of a 600 m^3^ tank.

**Figure 2 materials-15-00225-f002:**
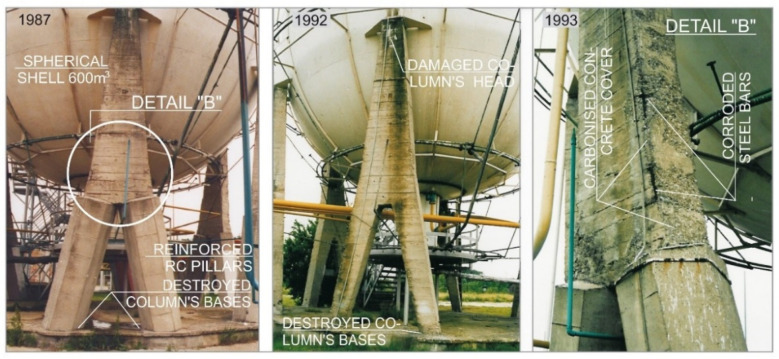
Structural faults of support structures of tanks with a capacity of 600 m^3^.

**Figure 3 materials-15-00225-f003:**
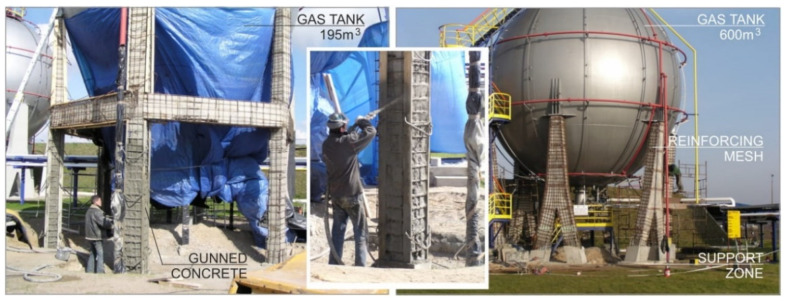
Details of the implemented tank-reinforcement structure.

**Figure 4 materials-15-00225-f004:**
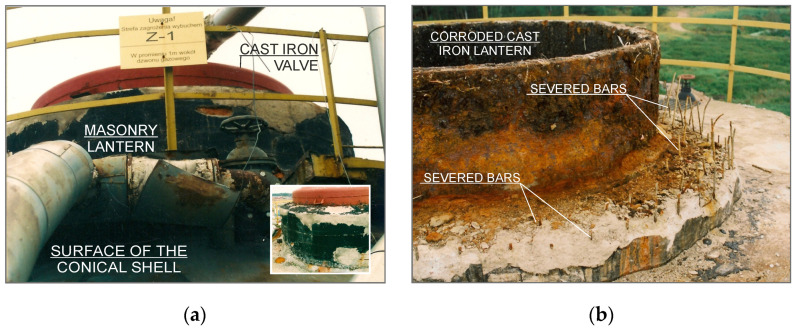
Degraded zones of fermentation chamber roofing shell: (**a**) before explosion; (**b**) after explosion.

**Figure 5 materials-15-00225-f005:**
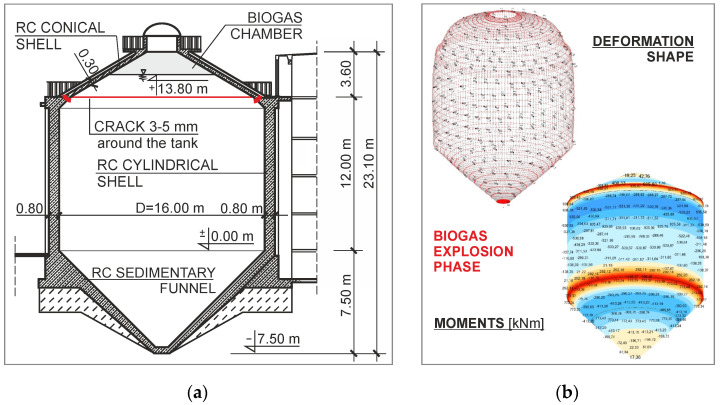
Digestive sewage tank: (**a**) cross section; (**b**) shape deformation and bending moments resulting from the explosion.

**Figure 6 materials-15-00225-f006:**
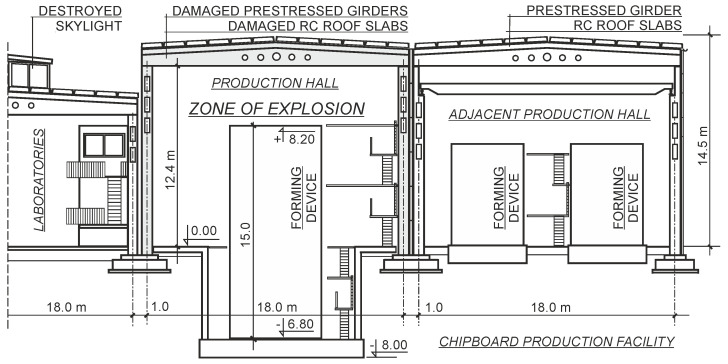
Cross−section of the damaged production hall.

**Figure 7 materials-15-00225-f007:**
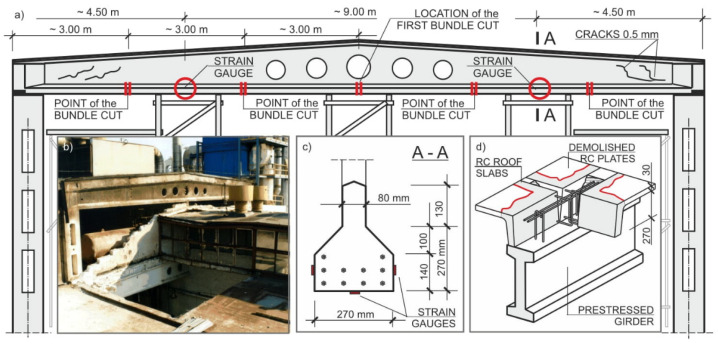
Production hall of wooden chipboards: (**a**) cross section; (**b**) degradation after explosion; (**c**) strain gauges’ locations, (**d**) RC plate demolition.

**Figure 8 materials-15-00225-f008:**
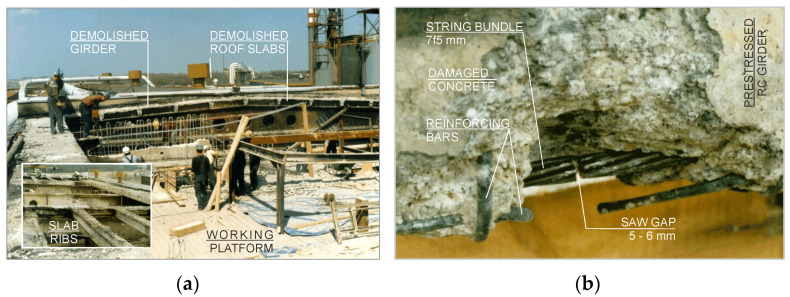
Demolition works of prestressed girders: (**a**) view of the working platform; (**b**) string cutting.

**Figure 9 materials-15-00225-f009:**
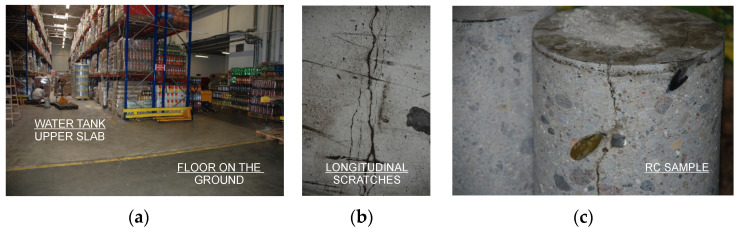
Shopping center warehouse: (**a**) slab-loading racks; (**b**) cracking of the top plate; (**c**) RC samples.

## Data Availability

The data presented in this study are available on request from the corresponding author.

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
