# Peer review of "Assessment of Destructive Impact of Different Factors on Concrete Structures Durability"

_materials, 2021, doi:10.3390/ma15010225_

Round 1

Reviewer 1 Report

The aim of this work is to discuss the problems caused by structural failures in concrete that affects their durability. Authors performed a good work of literature review addressing these problems, that are well described in the text properly discussed.

In the discussion, one point that could be improved in the text is the inclusion of standard standards (ASTM or others) associated with the constructions discussed and what they recommend, in terms of testing and durability, including aggressive environments (acidic or other media)

Because civil constructions are always carried out based on these standards. Adverse situations, such as explosions, premature aging, can happen, however, the discussion of failures can include the recommendations of standard norms. Therefore, if the authors include this discussion, the article will be improved. If there are no standardized norms, I believe this information is also relevant.

Otherwise, the work is well written, well discussed, the literature is appropriate and current, and, in my opinion, meets the standards of Materials journal

The article size and figures are adequate. In relation to the figures, the Editor may be able to question the authorship or copyright for publication, if this was not done when submitting the work.

Author Response

Point 1. The aim of this work is to discuss the problems caused by structural failures in concrete that affects their durability. Authors performed a good work of literature review addressing these problems, that are well described in the text properly discussed.

Response 1. Thank you very much for your very accurate comments, which helped me improve the manuscript.

Point 2. In the discussion, one point that could be improved in the text is the inclusion of standard standards (ASTM or others) associated with the constructions discussed and what they recommend, in terms of testing and durability, including aggressive environments (acidic or other media).

Because civil constructions are always carried out based on these standards. Adverse situations, such as explosions, premature aging, can happen, however, the discussion of failures can include the recommendations of standard norms. Therefore, if the authors include this discussion, the article will be improved. If there are no standardized norms, I believe this information is also relevant.

Response 2. In accordance with the Reviewer’s recommendation, I have supplemented the text with explanations regarding the applicability of standard guidelines in the field of diagnostic tests of structures operating in an aggressive environment and hazardous conditions. I also indicated the necessity to use methods based on scientific experience for diagnostic tests. Regarding the climate zones, I indicated that the objects that were tested by the author were located in Poland.

Point 3. Otherwise, the work is well written, well discussed, the literature is appropriate and current, and, in my opinion, meets the standards of Materials journal.

Response 3. Thank you for the comment.

Point 4. The article size and figures are adequate. In relation to the figures, the Editor may be able to question the authorship or copyright for publication, if this was not done when submitting the work.

Response 4. The author declares that he has the right to use the figures and the photographs prepared on his own.

Reviewer 2 Report

This paper presents a study of the effects of several factors on the safety and durability of concrete structure of some buildings, with the proposition of some restoration methods. The results of the presented research are valuable to building construction community.

However, before publication there are several aspects that should be addressed to provide readers with a better understanding of pertinent details used in this study and increase the value of the final conclusions:

  1. Line 98: an angle of approximately 6o to the vertical axis.
  2. You have studied some buildings; please indicate their age to have an idea.
  3. What is the concrete materials composition of C25/30?
  4. It is important to indicate the location and climate zone condition.
  5. Why using 8 mm of C25/30 and not more?
  6. Explain briefly the shortcreting spraying technology.
  7. The explosion of fermentation chamber is due to a flame or is due to accumulated gas?
  8. How did you know the deformed shape of exploded biogas tank (Figure 5b)?
  9. Can you provide some numerical measurement of strain gauges at the top of production hall (Figure 7)?
  10. The pillars used for reinforcing tank structure are similar to the tidal turbine reinforcement under the sea. You have talk about computational methods in the introduction, you can cite some works below:

- Laaouidi, H., Tarfaoui, M., Nachtane, M., Trihi, M., & Lagdani, O. (2020). A Comprehensive Numerical Investigation on the Mechanical Performance of Hybrid Composite Tidal Current Turbine under Accidental Impact. International Journal of Automotive and Mechanical Engineering17(4), 8338-8350.

- Nachtane, M., Tarfaoui, M., Goda, I., & Rouway, M. (2020). A review on the technologies, design considerations and numerical models of tidal current turbines. Renewable Energy157, 1274-1288.

Author Response

Point 1. This paper presents a study of the effects of several factors on the safety and durability of concrete structure of some buildings, with the proposition of some restoration methods. The results of the presented research are valuable to building construction community.

Response 1. Thank you very much for your very accurate comments, which helped me improve the manuscript.

Point 2. However, before publication there are several aspects that should be addressed to provide readers with a better understanding of pertinent details used in this study and increase the value of the final conclusions:

  1. Line 98: an angle of approximately 6o to the vertical axis.
  2. You have studied some buildings; please indicate their age to have an idea.
  3. What is the concrete materials composition of C25/30?
  4. It is important to indicate the location and climate zone condition.
  5. Why using 8 mm of C25/30 and not more?
  6. Explain briefly the shortcreting spraying technology.
  7. The explosion of fermentation chamber is due to a flame or is due to accumulated gas?
  8. How did you know the deformed shape of exploded biogas tank (Figure 5b)?
  9. Can you provide some numerical measurement of strain gauges at the top of production hall (Figure 7)?
  10. The pillars used for reinforcing tank structure are similar to the tidal turbine reinforcement under the sea. You have talk about computational methods in the introduction, you can cite some works below:

- Laaouidi, H., Tarfaoui, M., Nachtane, M., Trihi, M., & Lagdani, O. (2020). A Comprehensive Numerical Investigation on the Mechanical Performance of Hybrid Composite Tidal Current Turbine under Accidental Impact. International Journal of Automotive and Mechanical Engineering, 17(4), 8338-8350.

- Nachtane, M., Tarfaoui, M., Goda, I., & Rouway, M. (2020). A review on the technologies, design considerations and numerical models of tidal current turbines. Renewable Energy, 157, 1274-1288.

Response 2.

  1. I corrected the typo in line 98.
  2. I indicated the buildings’ erection dates.
  3. The information regarding concrete material composition was supplemented.
  4. I mentioned that all facilities were erected in central Europe climate zone, i.e., in Poland.
  5. It was explained that the maximum diameter of the reinforcing bars used was determined by the thickness of the reinforcing layer, limited by the elements of technological installations.
  6. The shortcreting spraying technology was explained.
  7. The text was supplemented by the statement that the explosion was initiated by an accidental introduction of a flame into the interior of the tank during renovation works in the area of the adjacent twin chamber, temporarily out of service.
  8. The results of numerical calculations verifying the post-disaster stage, which considered the change in the static pattern, allowed to determine the actual state of stresses and deformations in the concrete structure.
  9. The issue was explained by the statement that the readings of the strain gauges after cutting the fibers were stable, which confirmed the sufficient adhesion of the fibers to the cover layer, despite a drastic decrease in the strength of concrete in the prestressed girders.
  10. The suggested literature items and some additional ones dealing with the issues of numerical investigation and strengthening of RC structures have been analyzed and cited.

Reviewer 3 Report

I have reviewed the manuscript “Assessment of destructive impact of different factors on concrete structures durability” by Janusz R. Krentowski. The work is well-organised and needs some revisions to be published in Materials. Here are my comments.

  1. Revise some typos in the reference list. Improve the readability of Figure 5(b).

  1. Within the Introduction and text, add some references dealing with degradation and durability of RC structures such as:

Castaldo, P., Palazzo, B., & Mariniello, A. (2017). Effects of the axial force eccentricity on the time-variant structural reliability of aging rc cross-sections subjected to chloride-induced corrosion. Engineering Structures, 130, 261-274.

  1. Biondini, M. Vergani Deteriorating beam finite element for nonlinear analysis of concrete structures under corrosion Struct Infrastruct Eng, 11 (4) (2015), pp. 519-532

  1. Add some comments on future developments in relation to reliability assessment of these structures.

Author Response

Point 1. I have reviewed the manuscript “Assessment of destructive impact of different factors on concrete structures durability” by Janusz R. Krentowski. The work is well-organised and needs some revisions to be published in Materials. Here are my comments.

  1. Revise some typos in the reference list. Improve the readability of Figure 5(b).

Response 1. Thank you very much for your very accurate comments, which helped me improve the paper. I corrected the typos in the reference list. Fig. 5 was improved. By implementing the suggestions of Reviewer 3, the author also improved the readability of other figures.

Point 2. Within the Introduction and text, add some references dealing with degradation and durability of RC structures such as:

Castaldo, P., Palazzo, B., & Mariniello, A. (2017). Effects of the axial force eccentricity on the time-variant structural reliability of aging rc cross-sections subjected to chloride-induced corrosion. Engineering Structures, 130, 261-274.

Biondini, M. Vergani Deteriorating beam finite element for nonlinear analysis of concrete structures under corrosion Struct Infrastruct Eng, 11 (4) (2015), pp. 519-532

Response 2. The suggested literature items and some additional ones dealing with the issues of investigation and strengthening of RC structures have been analyzed and cited.

Point 3. Add some comments on future developments in relation to reliability assessment of these structures.

Response 3. I added to the Discussion section some recommendations regarding future assessment of the presented structures.

Round 2

Reviewer 2 Report

The authors have done an exceptional job responding to my comments and have made adequate revisions to their manuscript. I believe this manuscript is ready for publication.

Reviewer 3 Report

Comments to the Authors

I have reviewed the revised manuscript “Assessment of destructive impact of different factors on concrete structures durability” by Janusz R. Krentowski. The revised can be published in Materials.